# Genetic Regulation of Avian Testis Development

**DOI:** 10.3390/genes12091459

**Published:** 2021-09-21

**Authors:** Martin Andres Estermann, Andrew Thomas Major, Craig Allen Smith

**Affiliations:** Department of Anatomy and Developmental Biology, Monash Biomedicine Discovery Institute, Monash University, Clayton, VIC 3800, Australia; martin.estermann@monash.edu (M.A.E.); andrew.major@monash.edu (A.T.M.)

**Keywords:** chicken testis, sex determination, *DMRT1*, gonadal development, *PAX2*, *SOX9*, *AMH*

## Abstract

As in other vertebrates, avian testes are the site of spermatogenesis and androgen production. The paired testes of birds differentiate during embryogenesis, first marked by the development of pre-Sertoli cells in the gonadal primordium and their condensation into seminiferous cords. Germ cells become enclosed in these cords and enter mitotic arrest, while steroidogenic Leydig cells subsequently differentiate around the cords. This review describes our current understanding of avian testis development at the cell biology and genetic levels. Most of this knowledge has come from studies on the chicken embryo, though other species are increasingly being examined. In chicken, testis development is governed by the Z-chromosome-linked *DMRT1* gene, which directly or indirectly activates the male factors, *HEMGN*, *SOX9* and *AMH*. Recent single cell RNA-seq has defined cell lineage specification during chicken testis development, while comparative studies point to deep conservation of avian testis formation. Lastly, we identify areas of future research on the genetics of avian testis development.

## 1. Introduction

The gonads are fundamental to reproduction in animals. Among vertebrates, gonadal morphology is deeply conserved, as expected for organs that are essential for propagation of the species. The testes of males harbour the male germline and they secrete androgens that contribute to masculinisation of the soma. Understanding the cell biology and genetics of testis formation during embryogenesis is important for improving knowledge in diverse areas, from human sexual development to livestock breeding and species conservation. This review focusses on testis development in birds. Most current knowledge of avian testis biology has come from studies in the chicken, an agriculturally important species and a long-standing developmental model organism. Understanding avian sexual development is important for the poultry industry, which seeks methods of producing single sex bird lines [1]. Only females are required by the egg industry, while males are preferred by the meat industry. Male hatchlings are typically culled by the layer industry because they do not lay eggs [2]. Understanding how sex is determined in the chicken will lead to new and innovative ways of modulating sex in poultry. Furthermore, the chicken embryo is an instructive model for gonadal development. Given the conservation of testis morphology, studies in the chicken embryo have shed light on human testis development and its disorders [3,4]. The chicken is particularly suited to rapid functional analysis of gonadal genes. Embryos can be accessed *in ovo*, genetically manipulated and gonadal phenotypes observed up to several days later [5,6,7,8]. 

This review firstly considers the genetic mechanism regulating gonadal sex determination in birds, with emphasis on the testis. We then outline the developmental processes involved in testis formation in the chicken model; establishment of the gonadal primordium, differentiation of the Sertoli cell lineage from supporting cell progenitors, and testicular differentiation. We describe the principle factors involved in chicken testicular morphogenesis, and the importance of the gene, *DMRT1* (Doublesex and mab-3 Related Transcription factor 1). Testis formation in other avians is also considered. Lastly, future directions in the field of avian testis biology are considered.

## 2. Avian Sex Determination

All birds have a ZZ; ZW sex chromosome system, which dictates “sex determination” across tissues of the body. Male is the homogametic sex (ZZ), while the female is heterogametic (ZW) [9,10]. In chicken and most other birds, the Z and W are well differentiated; the Z is a large chromosome and the W is a smaller degraded version of the Z, with few *bona fide* genes [9,11,12,13,14]. Remarkably, the incidence of rare gynandromorphic chickens suggests that sex chromosomes have a direct effect upon sexual differentiation of the body, with a lesser role for hormonal signalling. Gynandromorphs are bilateral sex chimeras, male on one side of the body and female on the other. Such birds are rare, but they have been reported in chickens and some other birds [15,16]. In the few gynandromorphic chickens that have been described, the “male” side of the body is predominantly composed of ZZ cells, while the female side is predominantly ZW. The male side has large comb and wattle, legs spurs, male feathering and thicker breast muscle. The female side has smaller wattle, small or absent spurs, female feathering and lighter breast muscle [15]. The gonads of these birds reflect the relative proportions of ZZ or ZW cells; testes when ZZ cells are present, and ovaries when ZW cells are predominant [15,17]. Gynandromorphs are unlikely to be generated by aberrant endocrine signalling, because hormones would be expected to flow equally to both sides of the body. In their detailed analysis of three gynandromorphic chickens, Clinton and colleagues (2012) noted that gynandromorphs are best explained by direct genetic effects of the sex chromosomes in every cell, coining the term “Cell Autonomous Sex Identity” (CASI) [18]. This phenomenon is supported by manipulation of gonadal development in chicken embryos. When male Green Fluorescent Protein (GFP)-labelled presumptive gonadal tissue is grafted into a non-GFP female recipient embryo at embryonic day 2 (stage 14), the labelled cells differentiate into Sertoli cells (the so-called male supporting cell lineage), despite being in an ovarian environment [15]. Similarly, when female GFP-labelled cells are grafted into a male embryo, the labelled cells followed their own endogenous female fate [15]. In the chicken gonad, this intrinsic “sex identity” appears to be specific to the supporting cells but not cells of the gonadal epithelium, as epithelial fate can be modulated by estrogen, resulting in a thickened gonadal cortex, the site of folliculogenesis [6]. In fact, estrogen is required early in avian gonadal development [19]. The estrogen-synthesising enzyme, Aromatase, is expressed in female but not male embryonic gonads, and is both necessary and sufficient for ovary formation [20,21,22,23]. During embryonic testis formation, Aromatase expression must be inhibited. Most recently, it has been shown that genetically male (ZZ) chickens with monoallelic deletion of the Z-linked gene *DMRT1* develop ovaries instead of testes, but the rest of the body remains largely male [24]. This supports the concept of CASI, and it also confirms *DMRT1* is the master chicken testis determinant (discussed further below). 

Other sex-linked genes must underly CASI in different cells, as *DMRT1* is not expressed beyond the gonads. In this regard, it is noteworthy that, unlike therian mammals, birds lack global Z chromosome dosage compensation. As a result, on average, males (ZZ) have twice the level of Z-linked gene expression seen in females (ZW). This applies to various tissues tested in embryonic and adult chickens, and in other birds that have been examined [25,26,27,28,29,30,31,32]. Hence, different levels of different Z-linked genes may regulate sexually dimorphic anatomy in birds; the *DMRT1* gene in the gonads, and other Z-linked genes in other tissues (or female-biased genes expressed from the W sex chromosome) [33]. Developmentally, it has been proposed that gynandromorphs may arise due to failure of polar body extrusion during meiosis in the ZW ovum, and subsequent fertilization of the W and Z pronuclei by two Z-bearing sperm. This rare event would produce to an embryo with ZW diploid cells on one side of the body and ZZ cells on the other side [15].

## 3. Structure of the Adult Avian Testis

Unlike in therian mammals, avian testes are located deep in the body cavity, and hence spermatogenesis progresses at a temperature at or slightly above 37°C. In seasonal breeding birds, the testes increase dramatically in size, regulated via neuroendocrine response to photoperiod. In some species, testes increased in size by over 50% during the breeding season, controlled by the hypothalamic-pituitary axis [34]. The paired testes of adult birds comprise seminiferous tubules that house germ cells and supporting Sertoli cells, surrounded by steroidogenic Leydig cells. This is exemplified in the chicken *Gallus gallus domesticus* (Figure 1A) [35]. The avian testis grows substantially between hatching and sexual maturation, when semen is produced and birds copulate (up to 12 weeks of age) [36]. Prior to sexual maturation in the chicken, interstitial tissue (steroidogenic and mesenchymal cells) is extensive, while the seminiferous tubules comprise a single layer of cells along the basal lamina (Sertoli cells and pro-spermatogonia) [35]. The lumen is narrow or absent. As post-hatching development proceeds, there is a period of rapid cell proliferation within the tubules, such that they come to occupy a greater percentage of total testis volume. After this period of post-hatching proliferation, Sertoli cells become mitotically inactive and undergo maturation, characterized by a columnar shape, polarization and alignment along the basal lamina of the testis tubule [37]. In chicken and quail, this phase of Sertoli cells development is regulated by pituitary derived FSH and gonadal testosterone [38,39]. As in mammals, the post-embryonic Sertoli cells intimately surround the germ cells, generating an essential microenvironment for spermatogenesis. In the mature chicken, stages of spermatogenesis, ranging from spermatogonia to mature spermatozoa, are present in the mature tubules (Figure 1B) [35]. As in mammals, Sertoli cells are present around the germ cells, providing a niche for spermatogenesis via secreted signals. In the mature chicken testis, Sertoli cells express the marker, SOX9 (Figure 1C). Spermatogonia and Sertoli cells are arrayed along the basal lamina of the seminiferous tubule, as demarcated by nuclear *DMRT1* immunofluorescence (Figure 1D). Mature chicken Sertoli cells and germ cells have been isolated and cultured in vitro for functional analysis and, in the case of germ cells, for the generation of genetically modified embryos and birds [40,41,42,43,44,45,46]. 

## 4. Embryonic Development of the Testis in the Chicken Model

The chicken embryo is a widely used model of vertebrate embryogenesis. In the area of avian sex determination and gonadal sex differentiation, most of our knowledge has come from studies on the chicken embryo [3,47,48,49,50]. Figure 2A shows the histology of testicular morphogenesis in the chicken model. The total period of embryogenesis in the chicken is 21 days. As in mammals, the embryonic gonads are of mesodermal origin, forming on the ventromedial surface of the mesonephric kidneys. In chicken, the first histological sign of gonad formation is emergence of the genital ridge, a local thickening of the coelomic epithelium overlying the mesonephros and an accumulation of mesenchyme immediately beneath the epithelium [51]. This occurs between embryonic days (E) 3.5 and 4.5, corresponding to Hamilton and Hamburger (HH) stages 21–25 [52]. Even earlier in development, at E2.0 (HH) cell proliferation in the medial lateral plate mesoderm of the chicken embryo generates gonadal precursor cells via local Sonic Hedgehog action, mediated via BMP4 [7,53]. The gonads at this stage are considered “indifferent” or “bipotential”, though their fate is already determined by their sex chromosome constitution (ZZ or ZW). Up to E5.5 (stage 27), the undifferentiated gonad comprised the outer coelomic epithelium and underlying cords of cells (the so-called “medullary cords”) interspersed with loose mesenchymal cells (Figure 2A). The left gonad (in both sexes) is somewhat larger than the right, and it accumulates a larger number of germ cells [50,54]. As development proceeds, this left-right gonadal asymmetry becomes very marked in females, but is reduced in males. Gonadal asymmetry is driven by left-biased action of the *PITX2* gene in chicken. Early genetic markers of the gonad prior to sexual differentiation include the transcription factor, LHX1, and the signalling molecule, *FGF9*, expressed in the gonadal surface epithelium, and SF1 (Ad4BP) and *PAX2*, expressed in the underlying mesenchyme [55,56]. The transcription factor, GATA4, is expressed in the outer epithelium and underlying medulla. In the mouse model, *FGF9* is required for proper testis formation, responding to the transcription factor, Sox9 [57,58]. In chicken, over-expression of *FGF9* stimulates cell proliferation and produces an enlarged gonad [55]. At least some of this *FGF9* derives from the adjacent mesonephric kidney in chicken [55]. As in mouse, *LHX1* is also likely to be required for early somatic cell proliferation in the avian gonad. Steroidogenic factor 1 (SF1, or Ad4BP) is strongly expressed in the mesenchyme of the chicken gonadal (and adrenal) primordium. SF1, encoded by the gene *Nr5a1* is essential for proper gonad and adrenal formation in mammals. *Nr5a1* null mice show gonadal and adrenal agenesis [59]. However, SF1 has not been functionally analysed during avian gonadogenesis. 

Sexually dimorphic gene expression is already apparent in morphologically undifferentiated chicken gonads at E4.5 (HH stage 25). This is primarily W-linked expression in females and elevated Z-linked expression in males due to incomplete dosage compensation [12,60]. There is currently no evidence that W-linked genes play a role in avian gonadal sex determination [61,62]. The W sex chromosome in chicken is a degenerate homologue of the Z chromosome, the latter carrying around 1000 genes. Those few *bona fide* genes present on the largely heterochromatic chicken W are likely to be important genes that are dosage sensitive and have not been lost over evolution [12,63,64]. Most Z-linked genes are more highly expressed in males (ZZ) and they encode proteins with general metabolic and cellular maintenance functions, un-related to sex [65,66]. However, at least two Z-linked are expressed in the gonads and implicated in testis development in chicken, *DMRT1* and *HEMGN* [67,68]. 

By embryonic day 5 (HH stage 25) in chicken, the undifferentiated gonadal primordium comprises a distinct medulla and a columnar or pseudostratified outer surface (coelomic) epithelium (Figure 2A) [69,70]. (Then latter is also called the germinal epithelium.) Primordial germ cells (PGCs), having migrated from the germinal crescent via the blood stream [42], populate both medulla and gonadal surface epithelium (Figure 2A and B). The germ cells express *DAZL* and *CVH* (Vasa), both of which encode RNA-binding factors [71]. In the chicken, as in mammals, gonadal sex differentiation occurs during embryonic life and involves sexually dimorphic gene expression [3,50,72]. The first overt morphological sign of testis formation is the enlargement and proliferation of somatic cells in the medullary cords of the gonad (Figure 2A) [51,69]. These cells represent the pre-Sertoli cell lineage. These cells become organized in the medullary epithelial-like cords, they elaborate basement membrane and enclose PGCs. In the chicken embryo, this occurs from E6.0 (HH stage 30 -31) [73]. Outside these medullary cords, desmin and fibronectin positive mesenchymal cells give rise to interstitial cells [74], the site of later myoid and steroidogenic Leydig cell development. By E9 (HH stage 35), testis (seminiferous) cords are apparent, and loose mesenchymal cells are dispersed in the surrounding interstitium (Figure 2A). High magnification view of the developing chicken testis at this time point reveals Sertoli cells and PGCs within the testis cords (Figure 2B). As in mammals, Sertoli cells represent the first testicular lineage to differentiate in the chicken testis, expressing the markers *DMRT1*, *AMH* and SOX9 (Figure 2C,D) [3,75].

The chicken gonad develops in close association with the mesonephric kidney (mesonephros). The contribution of the mesonephros to chicken gonad formation has been debated. In the mouse, lineage tracing and single cell RNA-sequencing have shown that cells derived from the mesonephros contribute to the interstitial steroidogenic and vascular lineages on the developing testis [76,77] (reviewed in [78]). In contrast, the key supporting cell lineage (Sertoli cells in the testis, pre-granulosa cells in ovary) derives from the coelomic epithelium in the mouse [79,80,81]. An intact mesonephros is apparently not required for chicken gonadal development; early ablation of the mesonephros does not perturb gonadal development and sexual differentiation, although gonads are smaller than normal [82,83]. However, mesonephric ablation in the chick still leaves a pool of nephrogenic mesenchyme, which can contribute to the gonad. 

## 5. Molecular Regulation of Avian Testis Formation and the Role of *DMRT1*

Most current knowledge of avian testis development has come from studies on the chicken embryo [5,84,85]. In contrast to the mouse, the key supporting cell lineage does not arise from the coelomic epithelium in the chicken embryo. Lineage tracing and single cell RNA-seq in the chicken have shown that the gonadal surface epithelium (coelomic epithelium) generates non-steroidogenic interstitial cells, while the supporting cells (Sertoli cells in the testis, pre-granulosa cells in the ovary) derive from nephrogenic mesenchyme [5,86]. In both sexes, single cell RNA-seq identifies the gonadal mesenchymal supporting cell progenitors as a population expressing the transcription factors, *PAX2*, OSR1, and *DMRT1* together with the WNT4 signalling molecule (Figure 3) [86]. *PAX2* is a Paired Box homeodomain protein. It has not previously been linked to vertebrate gonadal development, but we find that it is expressed in the gonadal medulla of the undifferentiated chicken gonad, with expression being down-regulated as the gonads commence sexual differentiation [86]. *OSR1* (Odd-Skipped Related Transcription Factor 1) is required for early specification of the urogenital system [87,88,89], but a later role in gonadal development *per se* has not been reported. In the chicken, *OSR1* expression is maintained in the female supporting cell lineage (pre-granulosa cells of the medulla) but expression declines in the testis. Similarly, WNT4 expression is maintained in the developing chicken ovary, but is down-regulated during testicular morphogenesis [86,90]. Of these four diagnostic factors expressed in the indifferent embryonic chicken gonad, *DMRT1* expression is maintained at a high level in males, and is more lowly expressed in females. *DMRT1* encodes a zinc finger-like transcription factor with a DNA-binding DM domain. This gene is now proven to be the master genetic trigger for testis formation in the chicken [24,67], and likely in all birds [61]. A recent comparative study conducted in our lab found that *PAX2* and *DMRT1* exhibit conserved expression in the embryonic gonads of Japanese quail (*Coturnix japonica*), zebra finch (*Taeniopygia guttata*) and emu (*Dromaius novaehollandiae*). These species represent each of the major clades of avians (Galliformes, Neoaves and Paleaognaths) (Estermann et al. in press). In each species, *PAX2* expression is extinguished as *DMRT1* expression is activated. As in chicken, *DMRT1* is more highly expressed in male vs female gonads in quail, finch and emu.

Elevated *DMRT1* expression in ZZ chicken embryos operates as the genetic switch for testis formation. The *DMRT1* gene is a deeply conserved among vertebrates, with a conserved role in testis development and function across all major clades [91,92,93,94]. In the mouse, *DMRT1* is required to maintain the postnatal Sertoli cell phenotype, and for regulating gametogenesis [95,96,97,98,99]. *DMRT1* is required for testis development in turtles with either “genotypic” or temperature-dependent sex determination [100,101,102]. Furthermore, *DMRT1* or paralogous genes are required for gonadal development in amphibians and fishes [103,104,105,106,107,108]. In these groups, *DMRT1* may act as primary testis-determining switch (e.g., some fishes and reptiles) or as a downstream player in Sertoli cell development and maintenance (e.g., mouse). In birds, *DMRT1* is located on the Z sex chromosome, present in two copies in males (ZZ) and in one copy in females (ZW) [109]. This chromosomal location makes the gene a prime candidate master regulator of gonadal sex determination. The avian Z sex chromosome is deeply conserved and Z-linkage of *DMRT1* applies to all birds analysed, even the phylogenetically distinct flightless ratites. In emu and ostrich, the sex chromosomes are poorly differentiated and they exhibit extensive meiotic crossing-over, but, significantly, not in the region harboring Z-linked *DMRT1* [10,110,111]. 

In embryonic chicken gonads, *DMRT1* is expressed in the supporting cell lineage of both sexes (pre-Sertoli cells in males and fetal pre-granulosa cell progenitors in females). However, it is always more highly expressed in males, reflecting the lack of global Z inactivation [112,113,114]. This initial two-fold difference in *DMRT1* expression between the sexes is sufficient to drive testis as opposed to ovary formation. Knockdown of *DMRT1* in ZZ chicken embryos using virally delivered short hairpin RNAs results in feminization of the gonads, with down-regulation of SOX9 and *AMH*, and upregulation of the ovarian markers, FOXL2 and Aromatase [67]. Conversely, mis-expression of *DMRT1* in ZW gonads by *in ovo* electroporation results in ectopic activation of SOX9 and *AMH*, and disruption of Aromatase expression [75]. Most recently, CRISPR/Cas9 genome editing was used to delete one copy of *DMRT1* in the chicken, causing ZZ birds to develop feminised gonads or ovaries. Two different studies have been reported. Ioannidis and colleagues used single stranded oligonucleotides (ssODNs) to introduce a loss-of-function mutation and monoallelic inactivation of *DMRT1*. This resulted in clear ovary development in genetically male (ZZ) birds (embryos and adults) [24]. Lee *et al*. used a Cas9-mediated NHEJ (Non-Homologous End Joining) to generate a disrupted *DMRT1* allele via GFP insertion. They found feminization but not complete ovary formation in *DMRT1*^-/Z^ birds, perhaps due to some off-targeting effects [115]. Nevertheless, overall, current data confirm that *DMRT1* is the master Z-linked genetic trigger for testis formation in the chicken [24,115]. Loss of *DMRT1* expression is dispensable for chicken ovary development, though oogenesis is impacted. Based on these studies, it is currently considered that elevated expression of *DMRT1* in ZZ avian embryos antagonizes the FOXL2-Aromatase axis in medullary cords, which otherwise results in estrogen synthesis and ovarian differentiation [24]. However, when one copy of *DMRT1* is deleted in ZZ birds, the rest of the body remains “male”, with male type weight, comb and wattle, supporting the notion of cell autonomous sex identity, as discussed above [24].

Testis-enriched *DMRT1* expression in conserved during gonadal development in other avian species. We have recently reported male up-regulation at the onset of gonadal sex differentiation in representatives of each major avian clades (Galliformes, Neoaves and Palaeognaths). As in the chicken, *DMRT1* is male up-regulated during embryonic gonadal development in all other avians examined, including diverse species such as the Japanese quail, zebra finch and emu, (Estermann et al., in press) [61]. The same applies to embryonic development of the gonads in the Muscovy duck [116]. These data strongly indicate that Z-linked *DMRT1* is the master regulator of testis formation across all birds, in line with its deep conservation and role in the testis of other animals. In birds, *DMRT1* may have been one of the first genes to become differentially expressed between the sexes early in the evolution of the Z and W sex chromosomes. It is expected that other male-specific genes have also been recruited and enriched on the avian Z [117,118], in light of the CASI concept. Through the analysis of 5’ regulatory regions, it has been proposed that *DMRT1* first played a role in germ cell development in vertebrates (fishes), but then became recruited to the somatic cells of the gonads, where it acquired a role in Sertoli cell specification as well [107]. This would apply to birds and reptiles, but in mammals, the early role of *DMRT1* has been supplanted by the Y-chromosome-linked *SRY* gene, at least among eutherians. 

In the embryonic chicken gonad, *DMRT1* is also expressed in the germline. Nuclear expression of the protein can be detected in chicken primordial germ cells from early stages of gonadal sex differentiation [119]. Its role in avian germ cells has not been fully explored, but it may play a role in directing male germ cell differentiation, from mitotic arrest to specification of spermatogonia. In the mouse, *DMRT1* plays multiple crucial roles in germ cell differentiation, in both sexes [92,120,121]. In the testis, it promotes mitotic arrest and suppresses pluripotency genes at embryonic stages and, in certain strains, is a tumor suppressor [122,123]. Postnatally, *DMRT1* is required in the mouse for resumption of germ cell mitosis, migration to the germ cell niche and survival of spermatogonia [95,98]. At adult stages in mouse, *DMRT1* regulates entry into meiosis [124]. It is likely that *DMRT1* has a comparable function in the avian germline. In chicken embryos that have been CRISPR/Cas9 edited to delete *DMRT1*, germ cells are present at embryonic stages and they populate the developing gonads as normal, but postnatal gametogenesis is disrupted [24].

## 6. The Sertoli Cell the Lineage: Genes Downstream of *DMRT1*

While the genetic trigger for testis formation differs among vertebrates, the downstream genetic hierarchy is at least partly conserved [125]. In birds, *DMRT1* engages the testis developmental pathway during early embryogenesis. Figure 4 show the temporal expression pattern of *DMRT1* together with other key testicular proteins, SOX9, *AMH* and Z-linked HEMOGEN (*HEMGN*). *DMRT1* is expressed first, from at least as early as E4.5 (stage 25). *HEMGN* is subsequently detected, at varying levels across cells, from E5.5 (stage 28), together with SOX9, which is more homogenously expressed. This is followed by *AMH* protein expression (Figure 4). SOX9 is a central hub gene required for the initiation of pre-Sertoli cell development in the gonadal medulla. SOX9 is a SOX box transcription factor in the same family as SRY. In mouse or human, the *SOX9* gene is activated by the concerted action of SRY and SF1, which bind the *SOX9* regulatory region [126,127,128,129,130]. In chicken, quail and duck, *SOX9* is male-up-regulated during testis formation [131,132,133]. Indirect evidence indicates that *DMRT1* activates *SOX9* during pre-Sertoli cell specification. SOX9 expression is down-regulated following *DMRT1* knockdown in male gonads (ZZ), and is ectopically activated when *DMRT1* is mis-expressed in female gonads (ZW) [24,67,75]. As in other vertebrates, this is likely to trigger the differentiation of somatic medullary cord cells into the pre-Sertoli cell lineage [134]. In mammals, one of the principal targets of SOX9 is *AMH*, a diagnostic Sertoli cell glycoprotein responsible for Müllerian duct regression [135]. Accordingly, *Sox9* transcripts are first detected prior to *AMH* transcripts in the mouse embryo. However, the chicken is notably different; *AMH* mRNA is first expressed prior to *SOX9* mRNA (E4.5 vs E6.5), and in both sexes, though always more highly expressed in males [85,136,137]. (*AMH* is also expressed in female chicken because the right Müllerian duct also regresses, in line with regression of the right female gonad.) The data indicate that SOX9 does not activate *AMH* gene expression in the avian model, although it may maintain it. *AMH* may be activated by *DMRT1*, as *AMH* expression declines following *DMRT1* knockdown, though direct evidence is lacking. Another possibility is that, while *AMH* transcription is initiated prior to that of *SOX9*, its translation into protein may be delayed, occurring after that of *SOX9*, as suggested by the immunofluorescence shown in Figure 4.

Testis formation in the avian model may primarily depend upon the activation of *SOX9* by *DMRT1*, with SOX9 then setting in motion the male genetic circuitry and repressing the female pathway. This would be analogous to the transient yet essential role of Sry in activating *Sox9* in the mouse. Some known targets of Sox9 in the gonad that are required for proper testis development in the mouse embryo also apply to chicken. One of these is *Prostaglandin D synthase* (*PDG2*), activated by Sox9 in mouse. This enzyme produces Prostaglandin D2, which is required for feedback amplification of *Sox9* transcription, and the recruitment and organization of Sertoli cell progenitors [138,139,140]. PGD2 is also expressed male-specifically in the embryonic chicken gonad and can up-regulate *SOX9*, at least in gonadal explants [141]. Another well-established target of Sox9 in mouse is *FGF9*, which is required for proper Sertoli cell development and testis formation [57]. Activation of *FGF9* by Sox9 also involves a positive feedback mechanism, whereby *FGF9* binding to Fgfr2 is required for sustained Sox9 expression in mouse Sertoli cells and for their proliferation [58,142,143,144,145]. In mouse, *FGF9* (masculinizing) antagonizes Wnt4 (feminizing) [58]. Our global and single cell RNA-seq studies have not identified *FGF9* as a male up-regulated factor in embryonic chicken gonad, although other FGFs are expressed [12,86]. 

Comparative SOX9 ChIP-seq analysis has been performed on mouse and chicken embryonic gonads [146]. In the chicken, over 1000 enriched SOX9 ChIP peaks were identified from E7 (stage 32) male gonads, with some 263 genes are shared with those in an RNA-seq dataset [146]. Intriguingly, this study found only one SOX9 target gene shared between mouse and chicken—*AMH*. This suggests that the Sertoli cell gene regulatory network activated by SOX9 may differ between the two species. ChIP-seq for direct transcriptional targets of SOX9 has been performed on embryonic mouse and bovine gonads. In that study, a conserved so-called “Sertoli cell signature” was identified, comprising genomic regulatory region motifs that bound SOX9, *DMRT1* and the transcription factor GATA4 [147]. SOX9 in mouse and cow was found to exhibit direct transcriptional activation of target genes, and also sex-specific splicing of target transcripts. Among the conserved transcriptional targets included *AMH* and *FGF9*, as expected, *SOX9* itself and other *SOX* genes, *WNT*s, *BMP*s and *FOXL2* (putative negative regulation for the latter). This data points to co-regulation of target genes by SOX9 acting together with *DMRT1*. It was reported in this study that the Sertoli cell signature of *DMRT1*-SOX9-GATA motif enrichment conserved in mice and cattle also occurs across the orthologous genomic regions of other mammals and indeed in chicken [147]. 

The direct transcriptional targets of *DMRT1* during avian testis development are currently unknown. In mouse, genome wide analysis of *DMRT1* targets has been examined using combined CHIP-seq and RNA expression analysis of juvenile testes. Some 1400 proximal promoter regions are bound by mouse *DMRT1*, including the *DMRT1* gene itself. Mouse *DMRT1* protein is bifunctional, acting as an activator or repressor [148]. GO term enrichment showed *DMRT1* target genes encoding MAP kinases, FGF and IGFs, nuclear hormone receptors, cell cycle regulators and metalloproteinase factors, and other DM domain genes [148]. *DMRT1*, and other DM domain proteins, can function as heterodimers, and a core mouse *DMRT1* consensus binding site has been identified (ACA^A^/_T_ TGT ^T^/_A_) [149]. These findings may also be applicable in chicken, but empirical *DMRT1*-ChIP experiments have not yet been reported for birds. In their study of *DMRT1*-edited chicken embryos, Lee and colleagues used RNA-seq of knock out gonads compared to wild type gonads to identify global *DMRT1* targets (direct or indirect). They found modules of genes which showed expression changes in E6 embryos and in one week old hatchlings after monoallelic deletion of *DMRT1*, many of which were Z-linked, and some of which specified long non-coding RNAs [115]. Some genes showed a direct “linear” relationship to changes in *DMRT1* expression level, while other genes differentially expressed between the sexes responded differently to *DMRT1* loss. Some 213 genes did not change after *DMRT1* deletion, but were still sexually dimorphic in wild type gonads, suggesting that some gonadal sex-determining pathways may be independent of *DMRT1*. It would be worthwhile to compare the RNA-seq datasets derived from *DMRT1*-edited embryos [115] with the chicken SOX9 ChIP dataset [147] to determine whether *DMRT1* and SOX9 co-regulate a suite of target genes in chicken, as uncovered in mammals. As in mammals, *DMRT1* is thought to have a dual role in birds, promoting testis development in male (ZZ) embryos and repressing ovarian development in ZW (female embryos). The *DMRT1* protein is therefore likely to act as a transcriptional activator and a transcriptional repressor in the embryonic avian gonad. Consistent with this idea, mis-expression the gene in female gonads induces SOX9, *HEMGN* and *AMH* and represses Aromatase expression [75], while monoallelic deletion induces the converse [24]. 

*HEMOGN* has been shown to play a role in embryonic chicken testis development. This gene encodes a nuclear-localized transcription factor with a role in modulating hematopoiesis [150]. An expression-based screen revealed *HEMGN* to be expressed in male but not female embryonic chicken gonad, and in the key Sertoli cell lineage [68]. Like *DMRT1*, *HEMGN* is Z-linked in chicken, though its expression is initiated after that of *DMRT1*, as shown in Figure 4 [68,75]. Over-expression of *DMRT1* in embryonic female chicken gonads can induce *HEMGN* expression, further implying that it lies downstream of *DMRT1* [75]. Over-expression of *HEMGN* in female gonads can induce masculinization, with ectopic SOX9 expression and repression of the female markers, FOXL2 and Aromatase [68]. Interestingly, this experiment also up-regulates *DMRT1*, implying a positive feedback relationship between the two factors. Targeted knockdown or knockout of *HEMGN* has not been reported. However, curiously, *HEMGN* expression has not been detected in the embryonic gonads of other avians, such as zebra finch and emu [61]. The role of *HEMGN* in differentiation of the Sertoli cell lineage could be chicken (or Galliform)-specific.

## 7. Chicken Sertoli Cell Differentiation Revealed by Single Cell RNA-Seq

Most recently, single cell RNA-seq (scRNA-seq) has been used to provide insight into the gene regulatory network activated by *DMRT1* and the induction of cell lineage specification. Compared to females (ZW), higher expression of *DMRT1* in males (ZZ) induces rapid upregulation of thousands of genes that results in the differentiation of Sertoli cells, including known factors such as *AMH* and *SOX9*, and previously unknown factors [86]. The scRNA-seq indicates that in addition to these “typical” Sertoli cells, expressing higher levels of *DMRT1*, *SOX9* and *AMH*, a second type of Sertoli cells was identified in embryonic chicken testis. These “type 2” Sertoli cell express the same markers as typical Sertoli cells, but in lower levels [86]. At E10.5, these two populations appear in different testicular regions. The typical Sertoli cells are in the basal testicular cords, whereas the type 2 cells localize in the peripheral testicular cords. The type 2 Sertoli cells express low levels of mitochondrial genes and high levels of *GSTA2* and *CBR4* (Figure 5) [86]. These differences point to a metabolic difference among Sertoli cell types. Interestingly, stem cells tend to rely on glycolytic metabolism rather than mitochondrial oxidative respiration [151], suggesting that the type 2 cells could be a stem-like population of Sertoli cells. The two cell types may represent two maturational states of Sertoli cells. The apical supporting cells are the first to up-regulate Sertoli cell markers such as *AMH* and SOX9, suggesting an apical-basal wave of differentiation. Hence, this may reflect distinctive stages of Sertoli cell maturation, one more mature than the other. Further research is required to explore this dichotomy further.

## 8. Interstitial and Steroidogenic Cell Differentiation

The supporting cell population is the first cell lineage to differentiate in the embryonic gonads of vertebrates such as mammals and birds. In males, these cells form Sertoli cells. Signalling from Sertoli cells is thought to then direct the other uncommitted cells down the testicular developmental pathway [152,153]. In mouse, such signals include Desert Hedgehog (Dhh), important for fetal steroidogenic Leydig cell, germline development and peritubular myoid cell differentiation [154,155,156,157], and PDGF signaling involved in endothelial cell migration from the mesonephros. The latter is required for proper seminiferous cord partitioning in mouse [77,158]. Male-specific immigration of these endothelial cell precursors in mouse is also observed in chicken and also involves PDGF signalling [159]. Migrating endothelial cells form a testis-specific coelomic blood vessel in the mouse, a process blocked by factors such as Wnt4 and follistatin in the developing ovary [160,161]. Although male-specific cell endothelial cell migration is also observed in the avian model (chicken) [159], no male specific blood vessel has been observed. Early morphological studies conducted in the 1950’s showed that vascularisation of the embryonic chicken gonad in both sexes commences from E5 (stage 28) [162]. In terms of *DHH* singaling, expression of this factor has not been detected in the embryonic chicken testis [12,86], at least up to E10.5. Hence some of the Sertoli-derived signals may differ between mammals and birds. 

Differentiation of the steroidogenic cell population (Leydig cells) is a critical step in testicular morphogenesis, being essential for androgen production and masculinisation. Histologically, fibroblastic-like Leydig cells containing lipid vesicles are first recognised in the chicken embryo at E8 (stage 32) [163], after the onset of Sertoli cell development and testis cord organisation. This also applies to other avians that have been examined, such as quail and ostrich [164,165]. The origin of Leydig cells appears to differ between the mouse and chicken models. In the mouse embryo, the steroidogenic lineage (fetal Leydig cells) largely derives from the same progenitor pool of coelomic epithelial cells that also gives rise to the pre-Sertoli cell lineage [80,81]. The latter has an *Sry/Sox9/Wt1/*DMRT1*/Gata4* molecular signature, while the Leydig precursors have a *Wt1/Gata4/Sf1* signature that features down-regulation of *Wt1* [78,166,167]. A second source of fetal Leydig cells in mouse is the mesonephric kidney, a population that appears not to be induced by Hedgehog signalling [168]. However, in the chicken model, single cell RNA-seq suggests that the steroidogenic lineage derives directly from the differentiating supporting cell lineage (Sertoli cells in males and the medullary cord/embryonic pre-granulosa cells in females) [86,169]. In the male chicken embryo, Sertoli cells are marked by expression of *DMRT, SOX9* and *AMH*. Between E6.5 and E10.5, steroidogenic embryonic Leydig cells appear to differentiate directly from a subset of Sertoli cells by a process that involves a sequential upregulation of steroidogenic markers (e.g., *CYP17A1*) and downregulation of Sertoli markers (Figure 6) [86]. At E8.5 (stage 34), three different cell populations are detected in the testis: Sertoli cells expressing *AMH* but not *CYP17A1*, Leydig cells expressing *CYP17A1* but not *AMH* and a small number of transitioning cells expressing both *AMH* and *CYP17A1* (Figure 6) [86]. Furthermore, in chicken, other steroidogenic enzymes are expressed in Sertoli cells, most notably, *3**β**-HSD*, suggesting that both supporting and steroidogenic cells play a role in the synthesis of androgens, at least during embryonic stages [170]. In the mouse, there is reciprocal cross-talk between the fetal Leydig and Sertoli cells, ensuring proper seminiferous cord form and function [171]. This is likely to also apply in the avian testis, though it has not been well explored. 

## 9. Sex Steroid Synthesis and Testis Development

In birds, estrogen production by the female gonadal medulla (embryonic pre-granulosa cells) is essential for proper ovary formation [19,22,23,172,173]. Aromatase enzyme, the rate-limiting factor in estrogen synthesis, is only expressed in female gonads at the onset of gonadal sex differentiation [21]. Evidence indicates that one of the main functions on *DMRT1* in genetic males is repression of aromatase expression, possibly via antagonism of the FOXL2 transcription factor [67]. However, a role for androgens in avian testis development, comparable to the role for estrogens for ovary development, is not apparent. Histochemical and endocrine assays conducted in the 1970’s and 1980’s indicated that the gonads of both sexes are steroidogenically active early, before the onset of gonadal sex differentiation. Based on early immunohistochemical studies, gonadal testosterone is first detectable prior to sexual differentiation, (as early as E3.5 (stage 19) in both sexes, becoming higher in ZZ males after E5.5 [174]. However, exogenously added testosterone does not masculinise embryonic chicken gonads [175].

More recent approaches such as RNA *in situ* hybridisation and RNA-seq have shown that the embryonic chicken testis expresses all enzymes required for testosterone synthesis from cholesterol (*CYP11A2*, *HSD3**β1**, CYP17A1* and *HSD17**β4**)* from the early stages of gonadal sex differentiation (at least as early as E6/stage 31) [12,20]. Based on one immunofluorescence and radioimmunoassay report, plasma testosterone was detected at E5.5 (stage 28) in the chicken, becoming higher in males from E7.5 (stage 32) [176]. However, some other studies using ELISA or immunoassay do not find sex differences in plasma testosterone levels at embryonic stages [177,178]. In females, androgen provides the substrate for aromatization and estrogen production when expression of *CYP19A1* (encoding aromatase) commences at E6/ stage 30. However, as noted above, there is no evidence that testosterone plays a decisive role in avian testicular morphogenesis [175]. Gonads in the chicken appear to come under pituitary regulation in the latter part of embryogenesis, when the hypothalamic-pituitary-gonadal axis is established (from E11-13) [179,180,181]. Prior to this time, chicken gonadal sex differentiation appears independent of pituitary gonadotrophins, based on hypophysectomy data [180,182].

## 10. Germ Cells in the Embryonic Chicken Testis

The primary role of the gonads is to facilitate gametogenesis [183]. As the somatic cells differentiate into distinct testicular or ovarian cell types during embryogenesis, the primordial germ cells (PGCs) are exposed to sex-specific signalling that affects their fate [184]. PGCs of the developing testis become enclosed in seminiferous cords, a conserved process seen in essentially all vertebrates. In mammals, a critical difference between testicular and ovarian germ cells is the timing of meiotic entry. While male germ cells remain arrested in mitosis during embryogenesis, female germ cells enter in meiosis, arresting in diplotene of meiosis I. In the mammalian embryo, several lines of evidence indicate that retinoic acid (RA) diffusing from the mesonephros induces germ cells to enter in meiosis in females [185,186], though this has recently been questioned [187,188]. In males, *CYP26B1* enzyme expressed in Sertoli cells is thought to degrade retinoic acid (RA), creating a physical barrier to meiosis in the adjacent PGCs [189]. Instead, spermatogonia are arrested in G1/G0 and do not enter meiosis until after birth. RA can also antagonise somatic development in the mouse testis, as *Cyp26b1* null mouse mutants have impaired steroidogenesis and mild ovotestis development [190]. In the chicken and other avians, PGCs arise at the anterior margin of the area pellucida, and in vitro data indicate that specification involves BMP4 [191]. The PGCs migrate through the germinal crescent and bloodstream to populate the gonads (more so in the left gonad) by E3.5 (see Figure 3) [42,192]. They express the diagnostic *CVH (VASA)* gene, which encodes an ATP-dependent RNA helicase [71]. While meiosis is initiated in female germ cells of the left gonadal cortex from E12.5 (based upon expression of the premeiotic marker, *STRA8*) [189], entry of male chicken germ cells into mitotic arrest is poorly documented. Embryonic chicken gonads express both the RA-synthesising enzyme, RALDH2, and the degrading enzyme, CYP26B1 [189], suggesting a role for RA in regulating meiosis, as in mouse. In embryonic chicken gonads cultured in vitro, exogenous RA can stimulate meiosis, an effect blunted by RALDH2 inhibition [193]. In the male chicken embryo, and unlike in mouse, Sertoli cells express both *RALDH2* and *CYP26B1* [189]. It seems curious that RA synthesis would occur in the seminiferous cords if the PGCs must be protected from RA in males. Hence, the co-expression of CYP26B1 to degrade the RA might be crucial. Alternatively, RA may not mediate induction of embryonic induction of meiosis in chicken, as has recently been debated in mouse [188]. Germ cell meiosis can be initiated in male chicken gonads cultured in the presence of the CYP26B1 inhibitor, ketoconazole [194], suggesting that either RA is involved in meiosis induction, or that CYP26B1 might metabolise another undefined substrate that opposes meiosis. In vitro data indicate that RA can promote PGC proliferation and that male chicken germ cells actually require RA to transition from the PGC stage into spermatogonial stem cells [191,195].

CVH immunostaining and flow cytometry have been used to study the dynamics of chicken embryonic germ cells [196,197]. Germ cell development in the embryonic chicken testis centres around cell cycle control and the meiosis versus mitosis decision. In E6 gonads, approximately 1200 PGCs are present in male and female embryonic chicken gonads. As gonadal sex differentiation proceeds, germ cells proliferate, but much more rapidly in females (24,000 in females vs 15,000 PGCs in males at E10 and 88,000 vs 17,000 by hatching) [196]. In the developing testis, embryonic PGC proliferation peaks around E14 (stage 38), at which time most germ cells are at G0/G1 of the cell cycle. By hatching, over 95% of male germ cells are at the G0/G1 phase, reflecting widespread mitotic arrest. The trigger for germ cell proliferation and subsequent arrest in the embryonic testis not firmly established, but it involves signalling from Sertoli cells. Studies in the mouse embryo implicate TGFβ super family members (TGFβ, Nodal, Activin A and BMPs) [198,199]. In the chicken embryo, germ cell proliferation and survival require FGF, and the MEK/ERK and PI3K/AKT intracellular signalling pathways (reviewed in [200]). Interestingly, our recent single cell RNA-seq data has revealed sexually dimorphic gene expression in chicken PGCs prior to mitotic arrest in males or meiosis entry in females [86]. This suggests that the differentiating gonad initiates specific male and female genetic programs earlier than detected by the cell cycle changes observed later in embryogenesis. In the mouse, RNA-seq has been applied to spermatogenesis, identifying novel stage-specific biomarkers and the developmental roadmap governing differentiation of the male germline [201,202]. Such approaches are yet to be applied to avian spermatogenesis. 

## 11. Interstitial Cells of the Embryonic Chicken Testis

Perhaps the most poorly understood cells of the embryonic testis are the non-steroidogenic interstitial cells. That is, those outside the seminiferous cords. These cells contribute to the peritubular myoid population, that surrounds the testis cords [203,204]. In the chicken embryo, these cells derive at least in part from the coelomic epithelium. When the coelomic epithelium is electroporated with an integrating GFP expression plasmid construct prior gonadogenesis at E2.5 (stage 14/15), labelled cells are found throughout the interstitium - and not in the testis cords - up to seven days later [86]. The interstitial cells express high levels of extracellular matrix genes, collagens, *DCN*, *POSTN* and *TCF21*. In E8.5 gonads, male interstitial cells can be differentiated from their female counterparts by up-regulation of *ACTA2* (α-Smooth Muscle Actin), suggesting this interstitial population to be also the source of peritubular myoid cells (Figure 5) [86]. The origin of these cells is the coelomic epithelium, via an epithelial to mesenchyme transition and ingression into the gonad form E2.5. In the male avian embryo, due to the absence of estrogens, the coelomic epithelial cells do not proliferate to form a cortex, as in females. After furnishing interstitial cell progenitors, the epithelial layer flattens progressively to form squamous like epithelium [16].

## 12. Summary and Future Directions

Avian testis development represents an ideal model for understanding the molecular genetics of vertebrate gonadal sex differentiation. Much of our knowledge in this area has come from studies on the chicken embryo. Figure 7 summarises testis development in the chicken and the central role of *DMRT1*. The early gonadal primordium forms under the inferred influence of the transcription factors SF1, WT1, GATA4, and the proven signalling molecules SHH, BMP and *FGF9*. In both genetic sexes, resident mesenchymal cells adopt a molecular signature comprising the transcription factors *DMRT1*, *OSR1*, *PAX2* and the signalling molecule WNT4. *PAX2* expression is conserved across other avians. Also, in all birds examined, Z-linked *DMRT1* expression is higher in males than in females. This higher expression (an initial two-fold dosage) is sufficient to drive specification of the pre-Sertoli cell lineage. In males, *PAX2*, OSR1 and WNT4 expression decline as *DMRT1* expression increases. *DMRT1* is required for SOX9, *HEMGN* and *AMH* expression, while inhibiting the FOXL2/ Aromatase female pathway. A subset of Sertoli cells up-regulate steroidogenic markers, giving rise the fetal Leydig cells. Signals from the Sertoli cell lineage must also drive interstitial cell development, and formation of the squamous surface epithelium, and induction of germ cell mitotic arrest (Figure 7A) though the exact nature of those signals is unknown. The resulting organ is a structurally and functionally integrated unit, supporting gametogenesis and male sex hormone production (Figure 7B).

Several questions around avian testis development remain to be answered. Of particular interest is how *DMRT1* is activated and how it initiates the testis developmental pathway. While the regulatory region required for *DMRT1* expression in rat Sertoli cells has been described [205,206], this data is lacking for chicken *DMRT1*. Based on expression co-localization, the GATA4 transcription factor is a likely common regulator [205]. However, in vivo and in vitro promoter and enhancer analysis needs to be conducted to empirically determine the regulatory region in chicken. The factors responsible for *DMRT1* transcriptional activation in chicken must be present in both sexes, but the outcome must be different based on Z sex chromosome dosage. It is intriguing to speculate how a two-fold dosage difference in *DMRT1* can direct testis vs ovary formation. This initial two-fold difference in expression must be amplified, as *DMRT1* expression levels become significantly elevated in males versus females as gonadal development proceeds [12,75], pointing to positive feedback of *DMRT1* on its own expression. *DMRT1* ChIP- seq experiments are required to identify the direct transcriptional targets of *DMRT1*, which are likely to be genes subjected to both positive and negative regulation. *SOX9, *HEMGN*,* and *AMH* are targets, as they respond to *DMRT1* manipulation, though this could be indirect. The clear importance of gene dosage suggests that *DMRT1* may interact with other proteins. The protein can form heterodimers with other DM domain factors in mammals [149], and may do so in chicken, or it may dimerise with other factors. The exact relationship between *DMRT1*, *HEMGN* and SOX9 also requires further analysis. What are the targets of *HEMGN*? Does it act together with *DMRT1*? 

Another area for research is epigenetics. At present, there is very little known about the epigenetic landscape of the embryonic chicken gonad. One study reported sex-specific epigenetic marks on the chicken *CYP19A1* (aromatase) gene (methylation and histone lysine methylation) and described how these marks are only partially modified when ZZ gonads are femininsed with estradiol [207]. This suggests that epigenetic signatures on key sex genes may be relatively stable in chicken. It will be of interest to examine the epigenetic marks on genes such as *DMRT1*, especially given the recent finding that temperature can induce histone modifications to the *DMRT1* locus as a molecular mechanism underling temperature-dependent sex determination in some reptiles. In the red-eared slider turtle, *Trachemys scripta*, thermosensitive H3K27 methylase, KDM6B, directly promotes the transcription of *DMRT1* by eliminating trimethylation of H3K27 near its promoter [208]. Histone and DNA methylation should be examined during gonadogenesis in avian species.

Lastly, knowledge of the interaction between Sertoli cells and germ cells will be important for poultry science and for saving endangered bird species. Most recently, methods of harvesting, storing and manipulating avian germ cells have been perfected [43,209,210] and methods of producing transgenic or knockout birds have been refined [24,115,211,212]. These methods have been built on the accumulated knowledge gathered on gonadal development and germ cell specification in birds. Researchers have successfully introduced germ cells into the embryonic gonads of chicken embryos that have been modified to lack all endogenous germ cells, greatly enhancing germline transmission [44]. This approach will be very useful for potentially propagating rare species or breeds in surrogate hosts and will have application to the global poultry industry. Hence, enhancing knowledge of avian testis formation and germ cell development will shed light on gonadogenesis more broadly and will have important direct application to avian biology.

## Figures and Tables

**Figure 1 genes-12-01459-f001:**
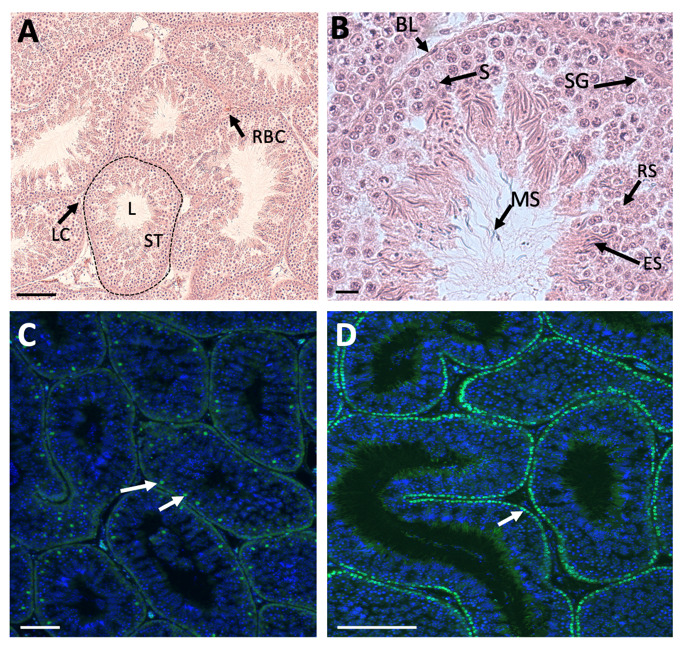
Histology and immunofluorescence of adult chicken testis. (**A**). Haematoxylin and eosin staining, showing seminiferous tubules (example: ST, dotted outline) and clusters of Leydig cells (example: LC). Red blood cells (RBC) are present between tubules. L, lumen of tubule. Scale bar = 100 µm. (**B**). High magnification view of adult testis tubule, showing Sertoli cells (example, S) and stages of spermatogenesis (example: SG, spermatogonia; RS, round spermatids; ES, elongated spermatids; BL, basal lamina; MS, mature spermatozoa). Scale bar = 10 µm. (**C**). Immunofluorescent localization of SOX9 (green) in the Sertoli cell nuclei of seminiferous tubules (example; white arrows). DAPI counterstain. Scale bar = 100 µm. (**D**). Immunofluorescent localization of *DMRT1* in the nuclei of spermatogonia and Sertoli cells (green), aligned along the basal lamina of the seminiferous tubule (example, white arrow). DAPI counterstain. Scale bar = 100 µm.

**Figure 2 genes-12-01459-f002:**
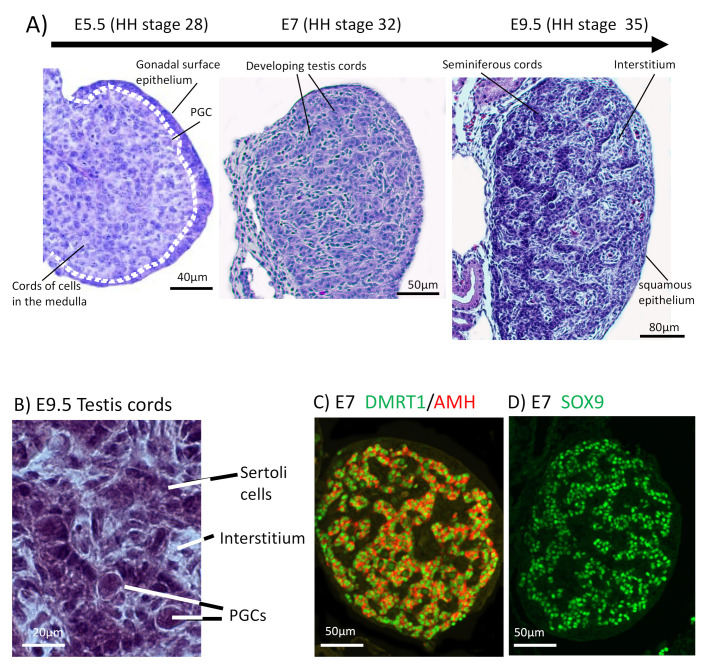
Testis development in the chicken embryo. (**A**) Histology of early testis differentiation, from E5.5 to E9.5. At E5.5, the gonad is morphologically undifferentiated, comprising outer epithelial cell layer and cords of cells in a dense underlying medulla. Dotted line demarcates the surface epithelium. Between E6.0 (stage 31) and E7.0 (stage 32), pre-Sertoli cells differentiate in the medulla, generating testis cords. By E9.5 (stage 35), Sertoli cells and primordial germ cells (PGCs) are apparent in testis (seminiferous) cords. The outer epithelium has become flattened, forming a squamous cell layer. (**B**) High magnification view of E9.5 (stage 35) testis cords, showing Sertoli cells, PGCs and surrounding mesenchymal cells in the interstitium. (**C**) Expression of the Sertoli cell markers, *DMRT1* (green) and *AMH* (red) in the developing testis cords at E7 (stage 32). (**D**) Expression of the Sertoli cell marker, SOX9 (green) in the nuclei of Sertoli cells at E7 (stage 35).

**Figure 3 genes-12-01459-f003:**
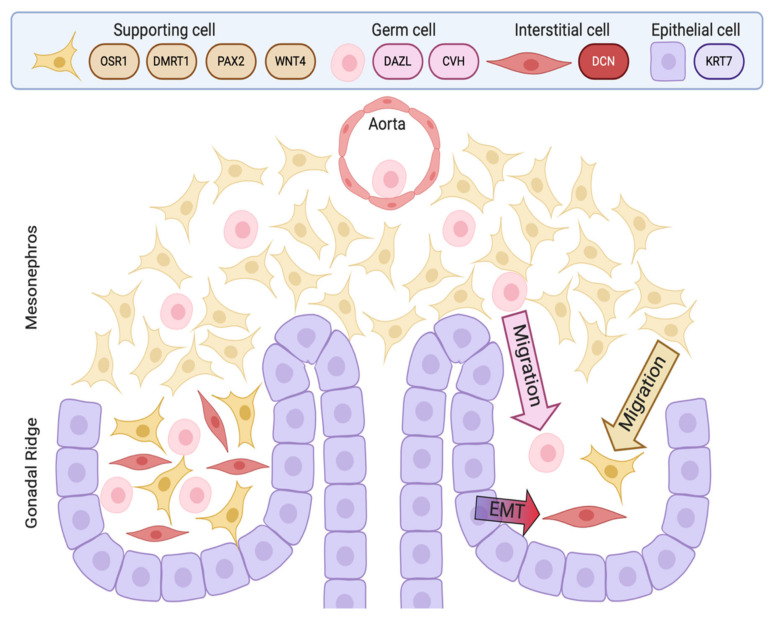
Schematic view of paired gonadal primordia development in the chicken embryo (E3.5- E4.5/stage 19-25). The supporting cell lineage (pre-Sertoli cells in males) derives from a resident mesenchymal cell population expressing *PAX2, OSR1, WNT4* and *DMRT1* (brown). The DAZL/CVH+ germ cells (pink) migrate into the gonad via the blood stream, exiting the dorsal aorta. The *Keratin 7*+ (KRT7) gonadal surface epithelium (coelomic epithelium) (purple), generates *Decorin* + (DCN) non-steroidogenic interstitial cells (red) via an Epithelial to Mesenchyme Transition (EMT), at least at early embryonic stages. image created with BioRender.

**Figure 4 genes-12-01459-f004:**
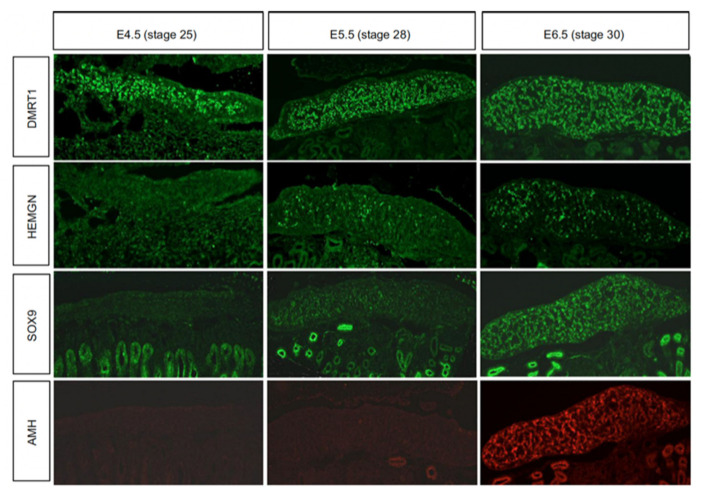
Chronology of marker protein expression in embryonic male chicken gonads at the onset of sexual differentiation.Immunofluorescence staining for *DMRT1* (green), *HEMGN* (green), SOX9 (green) and *AMH* (red) proteins in E4.5 (HH25), E5.5 (HH28) and E.6.5 (HH30) embryonic gonads. *DMRT1* shows robust expression from at least as early as E4.5. Both *HEMGN* and SOX9 are expressed from E.5.5 (stage 28), and *AMH* protein is detectable at E6.5 (stage 30-31). Scale bar = 100 µm. Reproduced from Lambeth et al., (2014) with permission.

**Figure 5 genes-12-01459-f005:**
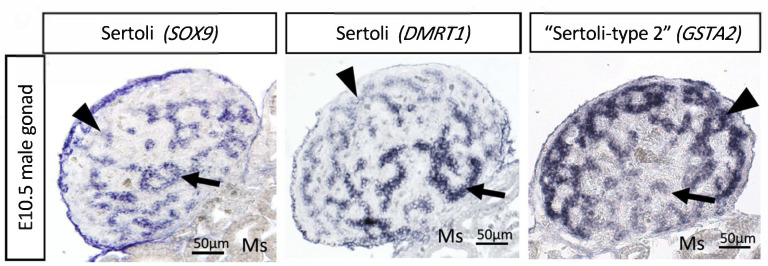
Marker gene expression in E10.5 (stage 35) embryonic chicken testis identifies two Sertoli cell populations. The first is more basally located and highly expresses SOX9 and *DMRT1* and lower level GSTA2 (arrows). The second (“type 2” Sertoli cells) is more peripherally located and highly expresses GSTA2 but shows lower levels of *DMRT1* and SOX9 expression. (Ms = mesonephric kidney). Reproduced form Estermann et al. (2020) with permission.

**Figure 6 genes-12-01459-f006:**
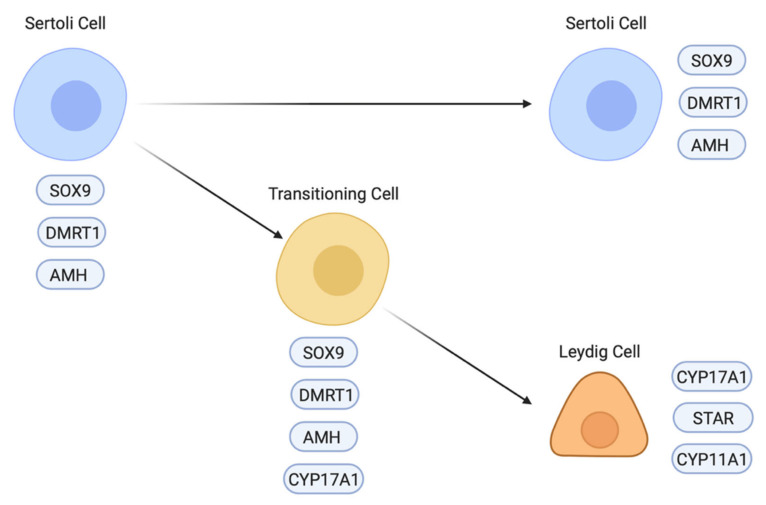
Leydig cells likely derive from a subpopulation of Sertoli cells in the chicken embryo. Some of the differentiated Sertoli cells (expressing *SOX9, *DMRT1** and *AMH*) start expressing the steroidogenic gene *CYP17A1* and start transitioning into a steroidogenic cell type. These transitioning cells express both Sertoli and Leydig cell markers. In order to differentiate into Leydig cells, the Sertoli cell markers are downregulated and steroidogenic cell markers (*StAR* and *CYP11A1*, etc.) are upregulated. Image created with BioRender.com.

**Figure 7 genes-12-01459-f007:**
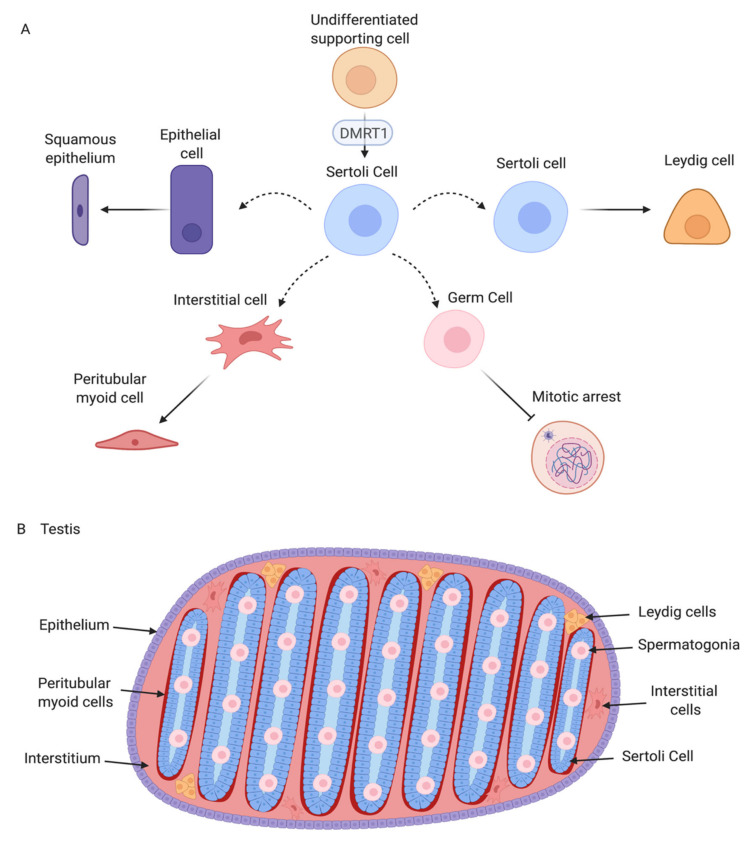
Testicular differentiation in the chicken embryo. (**A**) The first cells to differentiate are the Sertoli cells. They then regulate differentiation of other cell types in the testis. Epithelial cells become squamous, interstitial cells differentiate into peritubular myoid cells and upregulate *ACTA2*. Germ cells do not enter meiosis and are arrested in in G1/G0. Some Sertoli cells differentiate into steroidogenic Leydig cells. (**B**) The testis has a clear organised structure of seminiferous tubules, containing Sertoli cells and germ cells. Myoid cells are present surrounding those tubules. Leydig cells, and interstitial cells are located in the space between the tubules. Image created with BioRender.com.

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
