# Peer review of "Genetic Regulation of Avian Testis Development"

_genes, 2021, doi:10.3390/genes12091459_

Round 1

Reviewer 1 Report

The objective of the submission is to provide a review of gonadal development in birds.  Included are some comparisons with mammals.

  1. Provided is a comprehensive review.  It relies heavily on the chicken, which is understandable because that is what much of the literature is based on.  There is need for editing of the reference section.
  2. Commercial poultry involves populations selected specifically for meat or eggs.  I do not believe (l. 32-34) that half of the chicks in the meat arena are discarded.  In the egg sector, the males are discarded because they do not produce a product.  The statement in the text (which is not referenced) needs qualification to the egg industry or where it is the case for broilers.
  3. l. 109. Between hatching and sexual maturation (0-6 weeks of age) requires a definition of sexual maturation.  One may phenotypically separate, in some stocks, males from females via secondary sex characteristics, but I do not know of mating activity or semen production at 6 weeks.
  4. There is inconsistency in the source of the figures.
  5. l. 375-378; 400. When I checked reference #145, I missed where they cited the chicken.
  6. l. 541. birth or hatch?
  7. l. 574. is not and l. 580 suggests

Author Response

Response:

  1. References have been edited correctly.
  2. Reviewer is correct. Half of hatchlings are culled for egg industry, but not meat industry (although males are preferred). We have reworded this, and added a reference. See lines 32-34 of revised text.
  3. The Reviewer is correct. Apologies, 6 weeks applies to quails, so this is an error. Sexual maturity in chicken is longer. Sentence should read 0-12 weeks. Also, sexual maturity is production of semen, and ability to mate. We have corrected this and added a reference also. See lines 111-113 of revised text.
  4. We are unclear about this comment on inconsistency of source of figures. Some are our own, and some are sourced from other papers (with permission.)
  5. This is now ref # 147. The reviewer is referring to this paper:

Rahmoun M, et al (2017). In mammalian foetal testes, SOX9 regulates expression of its target genes by binding to genomic regions with conserved signatures. Nucleic Acids Res. 2017 Jul 7;45(12):7191-7211

The reviewer states that they missed any citation to chicken in this paper. This paper is indeed about mouse and cattle, but on page 7208l, paragraph 2, the authors state that the same Sertoli cell binding signature was seen in chicken (quoted as “not shown”).

  1. Birth is correct, as mouse is being discussed.
  2. Typos correct

Reviewer 2 Report

This is an outstanding review on the title topic.

Only few formal indications:

  • 1) Abstract line 13: "...., though other species ...."
  • 2) Line 73: define GFP-labelled
  • 3) Line 137: "....embryonic gonads ...."
  • 4) Line 196: "....pre-Sertoli, Sertoli cells ...."
  • 5) Line 200: eliminate "in"
  • 6) Line 280: "... is deleted in ...."
  • 7) Line 350: "... for proper testis development ..."
  • 8) Line 451: " .... PDGF signalling involved..."
  • 9) Line 486: " ...are expressed in Sertoli ...."
  • 10) Line 519: " .... or immunoassay ...."
  • 11) Line 542: " ... development in the mouse ..."
  • 12) Line 620: at the end, cite (Figure 7B)
  • 13) Line 626: "...co-localization, ..."

Author Response

Response:

Reviewer 2 requested several minor typos to be corrected. This has been done, as shown on the tracked changes.